# Growth Peculiarities and Properties of K$R_3$F$_{10}$ ($R$ = Y, Tb) Single Crystals

**Denis N. Karimov** [1,*], **Irina I. Buchinskaya** [1], **Natalia A. Arkharova** [1], **Anna G. Ivanova** [1], **Alexander G. Savelyev** [1], **Nikolay I. Sorokin** [1] **and Pavel A. Popov** [2]

1   Shubnikov Institute of Crystallography of Federal Scientific Research Center «Crystallography and Photonics», Russian Academy of Sciences, Leninskiy Prospekt 59, 119333 Moscow, Russia; buchinskayaii@gmail.com (I.I.B.); natalya.arkharova@yandex.ru (N.A.A.); ani.crys.ras@gmail.com (A.G.I.); a.g.savelyev@gmail.com (A.G.S.); nsorokin1@yandex.ru (N.I.S.)
2   Department of Physics and Mathematics, Petrovsky Bryansk State University, Bezhitskaya Str. 14, 241036 Bryansk, Russia; tfbgubry@mail.ru
*   Correspondence: dnkarimov@gmail.com; Tel.: +7-903-778-74-89

**Abstract:** Cubic K$R_3$F$_{10}$ ($R$ = Y, Tb) single crystals have been successfully grown using the Bridgman technique. Growth of crystals of this type is complicated due to the hygroscopicity of potassium fluoride and melt overheating. The solution to the problem of oxygen-incorporated impurities has been demonstrated through the utilization of potassium hydrofluoride as a precursor. In this study, the crystal quality, structure features, and optical, thermal and electrophysical properties of K$R_3$F$_{10}$ were examined. Data on the temperature dependences of conductivity properties of KTb$_3$F$_{10}$ crystals were obtained for the first time. These crystals indicated thermal conductivity equal to $1.54 \pm 0.05$ Wm$^{-1}$K$^{-1}$ at room temperature caused by strong phonon scattering in the Tb-based crystal lattice. Ionic conductivities of KY$_3$F$_{10}$ and KTb$_3$F$_{10}$ single crystals were $4.9 \times 10^{-8}$ and $1.2 \times 10^{-10}$ S/cm at 500 K, respectively, and the observed difference was determined by the activation enthalpy of F$^-$ ion migration. Comparison of the physical properties of the grown K$R_3$F$_{10}$ crystals with the closest crystalline analog from the family of Na$_{0.5-x}$R$_{0.5+x}$F$_{2+2x}$ ($R$ = Tb, Y) cubic solid solutions is reported.

**Keywords:** complex fluoride; growth from the melt; bulk crystals; KY$_3$F$_{10}$; KTb$_3$F$_{10}$; Bridgman technique; rare-earth ions; thermal conductivity; ionic conductivity; optical materials

## 1. Introduction

Crystalline materials are the basis of any optical and optoelectronic device. The possibility to grow bulk crystals with unique fundamental properties determines the functionality of the instrumentation, practical technological applications, and common progress in science and industry. Among complex inorganic fluorides based on rare-earth ions, K$R_3$F$_{10}$ (where $R$ = Y, Tb–Lu) single crystals with a cubic structure attract special attention as promising materials for optics and photonics [1–4].

KY$_3$F$_{10}$ (KYF) single crystals, as the best studied representatives of this large K$R_3$F$_{10}$ family, have acquired practical importance as a construction optical material and a laser matrix for various rare-earth ion doping [4–15]. Another prospective representative, KTb$_3$F$_{10}$ (KTF) crystal, is the next generation magneto-optical material for the visible and near infrared optical Faraday isolators [16–19]. In contrast to Tb-based fluoride crystals such as TbF$_3$ and LiTbF$_4$, this material is optically isotropic and has serious advantages over the traditionally used magneto-optical terbium-gallium garnet crystals due to their excellent thermo-optical performance. In addition, KTb$_3$F$_{10}$ crystals are of interest as a high power converter phosphor for LEDs and a potential candidate for X-ray slow scintillating applications [20].

For the first time, $KTb_3F_{10}$ single crystals were grown using the Czochralski method [21,22], and at present the industrial crystallization of these crystals is carried out by Northrop Grumman SYNOPTICS (USA) using a top-seeded solution growth method [23,24]. Recently, optimization of the melt composition using the Bridgman–Stockbarger method for $KTb_3F_{10}$ crystal growth was carried out, and the incongruent (peritectic) melting of this compound and the unambiguous presence of the homogeneity region were confirmed [25].

Pure and rare-earth doped $KY_3F_{10}$ single crystals are grown from a melt by various methods of directional crystallization [4,26–28]. These crystals have a congruent melting characteristic and melting point at $T = 1263$ K. A detailed review of phase interactions in the $KF–YF_3$ system was reported in [29,30]. The crystal structure of this fluoride is considered a cubic $2 \times 2 \times 2$ fluorite superstructure (space group $Fm\overline{3}m$, $Z = 8$), which consists of two ionic groups $[KY_3F_8]^{2+}$ and $[KY_3F_{12}]^{2-}$ alternating along the three crystallographic directions [27,28].

However, study of the growth process peculiarities and the characterization of $KR_3F_{10}$ bulk crystals have not been sufficiently conducted so far. These crystals are considered difficult to grow with high optical quality, even with well-established techniques. Compared to other fluoride compounds, $KR_3F_{10}$ crystals often grow cloudy. It is possible to obtain high quality of these crystals only using absolutely dry initial reagents and a pure growth process [1].

The aim of this paper is investigation of some features of the $KR_3F_{10}$ crystal growth by the vertical directional crystallization for $R = Y$ and Tb, possessing a distinctly congruent and incongruent melting characteristic, respectively, and investigation of physical properties.

## 2. Materials and Methods

### 2.1. Growth Equipment Design

At present, large-size fluoride single crystals are grown from the melt using the Czochralski and Bridgman (Stockbarger) methods [31–34], and less often using the Kyropoulos and vertical gradient freeze (VGF) methods. The micro-pulling-down (μ–PD) method for obtaining single-crystal fibers has also been presented [35,36]. The Czochralski method allows high quality single crystals to be obtained. However this technology is expensive and complicated in comparison with the Bridgman technique.

In this work, the Bridgman–Stockbarger method for growing $KR_3F_{10}$ single crystals was used. The use of a heating unit in the single-heater configuration to provide a sharp temperature gradient at the crystal–melt interface and in the cooling zone allows the growth of crystals of the simple fluorides and multicomponent congruently melting compositions. In most cases such thermal conditions lead to significant mechanical stresses and block the crystal structure. The growth of perfect crystals (especially incongruently melting and with pronounced cleavage) without plastic deformation traces requires a presence of the special zone with a relatively small temperature gradient at the cooling and annealing stages. Thus, complex multi-zone temperature conditions are required for the crystal growth. The best results can be obtained by using a configuration of multi-zone heating units (with two or more heaters) and a water-cooled diaphragm between them. Modeling of the crystallization conditions for our objects (transparent dielectrics) is complicated by the fact that the dominant thermal radiation in the process of heat transfer in the melt and crystal introduces nonlinearity. Complex numerical and experimental studies of thermal fields during crystallization of partially transparent materials have demonstrated that to maintain optimal crystallization conditions (uniform axial temperature gradient and preservation of the linear solidification rate), programmed variation of the heaters electric power is required [31,37].

The heating furnace split into two independent sections by means of Mo or a graphite diaphragm was employed in this work. (Figure 1a,b). This equipment allows crystals to be grown using various methods of vertical directional crystallization of the melt (Bridgman–Stockbarger, gradient freezing, Kyropoulos methods) under a minor reconstruction of the

heating unit. Thus, it is possible to grow crystals of high-temperature hydrolysable fluoride with diameter up to 80 mm and length up to 140 mm, both in vacuum and in a fluorinating atmosphere [38–41]. The upper temperature limit is 2250 K. The growth chamber provides deep evacuation to a residual pressure of $10^{-3}$ Pa using a turbomolecular pump system.

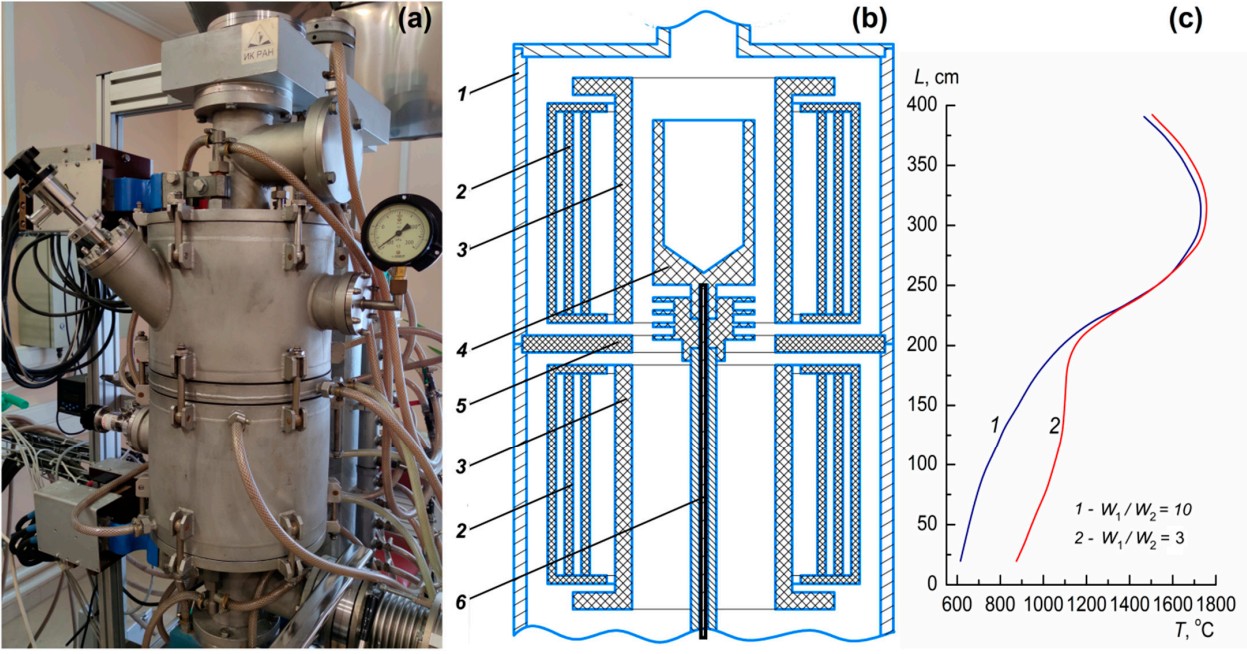

**Figure 1.** The external view of the two-section growth facility (**a**); a simplified design of the furnace heating unit: 1—water-cooled chamber shell; 2—independent blocks of the radiation thermal screens (graphite/ carbon felt); 3—upper and lower resistance heaters; 4—graphite crucible; 5—graphite (molybdenum) diaphragm; 6—pulling rod with crucible supporting holder and inner W/Re thermocouple (**b**) and axial temperature distribution along the growth chamber length at the different electric power ratios of the upper ($W_1$) and lower ($W_2$) resistance heaters (**c**).

The axial temperature gradient at the crystallization front is varied through the double-zone configuration of the heating unit (Figure 1c). Such a configuration provides fine tuning of the crystallization process thermal parameters for a specific type of crystal, both in the growth zone and in the cooling zones, and ultimately improves the quality of the grown crystals. By varying the electric power ratio ($W_1/W_2$) of the heaters, it is possible to change the temperature gradient in the range of 15–100 K/cm in the growth zone and create practically gradient-free conditions in the cooling zone under optimally selected parameters of the crystallization process.

*2.2. Initial Chemical Reagents and Growth Parameters of Growth Process*

As mentioned above, KYF crystals have a distinctly congruent melting character, whereas KTF crystals melts incongruently, and this greatly complicates the growth of this Tb-based compound (Figure 2). The initial melt composition for growing KYF is determined by its stoichiometry, although the optimal composition of the melt for KTF growth by the Bridgman–Stockbarger technique corresponds to a content of $27.5 \pm 0.5$ mol. % KF [25].

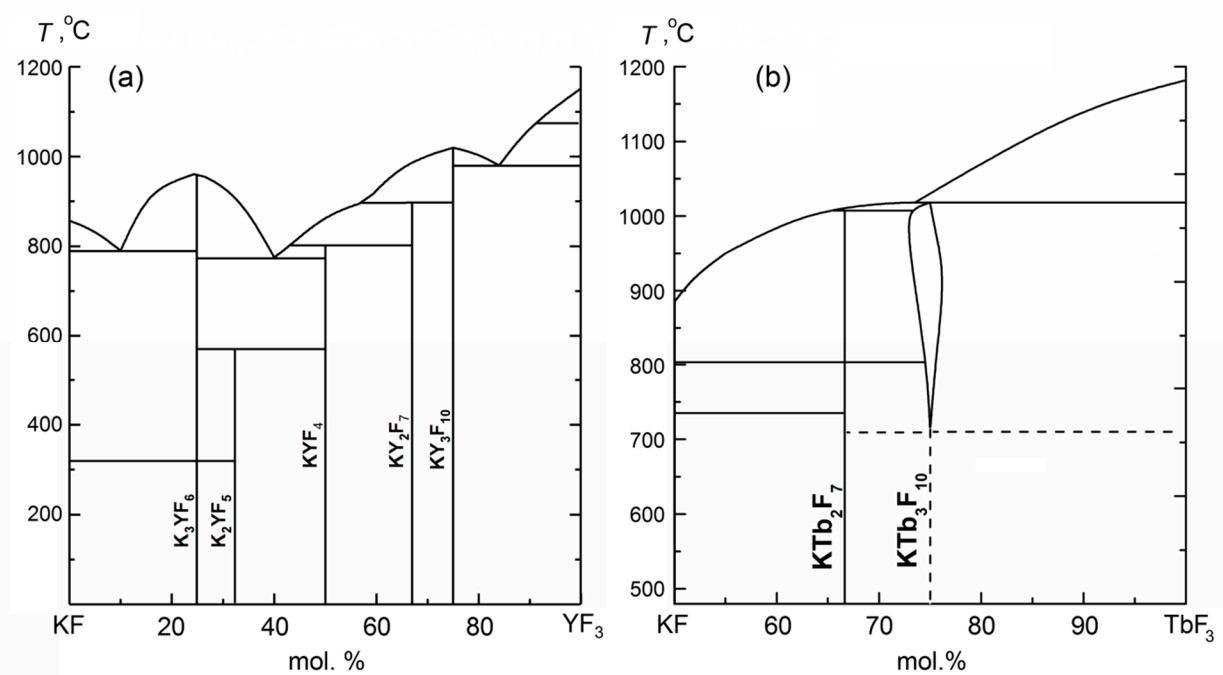

**Figure 2.** Phase diagrams of KF–YF$_3$ [29] (**a**) and KF–TbF$_3$ [25] (**b**) systems.

The high purity anhydrous powder YF$_3$, TbF$_3$ (99.99%, LANHIT, Moscow, Russia), KF ($\geq$99.9% Sigma-Aldrich, Louis, MO, USA), and laboratory-made hydrofluoride KHF$_2$, which was obtained by the interaction of carbonate K$_2$CO$_3$ (99.995%, Sigma-Aldrich) with a concentrated HF solution, were utilized as raw materials. The rare-earth trifluoride powders were preliminarily annealed in vacuum (~10$^{-2}$ Pa) for 3–5 h at 450 K, then melted in a fluorinating (He+CF$_4$+HF) atmosphere for deep purification from oxygen-containing impurities. Directional crystallization of KYF and KTF was carried out in a He+CF$_4$ atmosphere in a multicellular graphite crucible. Fused YF$_3$ or TbF$_3$ were separately placed in the crucible channels as seeds. The melt composition was stoichiometric (25/75 mol. %) for KYF and enriched in KF (28/72 mol. %) for KTF.

The following technological parameters of the growth process were applied: preliminary homogenization of the melt for 3 h, melt overheating of about 100 K, the temperature was controlled by the W/Re thermocouple and by the reference substance melting (TbF$_3$, $T_m$ = 1455 $\pm$ 8 K); the temperature gradient in the growth zone was ~80 K/cm. After starting the growth, the pulling rate was set to 2–3 mm/h; the cooling rate of the crystals was 50–100 K/h. The evaporation losses during the crystallization process did not exceed 1 wt. %. Thus, 30–50 mm long KR$_3$F$_{10}$ crystals with 10–30 mm in diameter were successfully grown.

Cubic Na$_{0.4}$R$_{0.6}$F$_{2.2}$ ($R$ = Y, Tb) single crystals were grown additionally according to previously described methods [42–44] for comparative analysis of the properties.

### 2.3. X-ray Diffraction (XRD) Analysis

The XRD analysis of the crystal was carried out using an X-ray powder diffractometer Rigaku MiniFlex 600 (CuK$\alpha$ radiation). The diffraction peaks were recorded within the angle range 2$\theta$ from 10 to 140$^\circ$. Crystal phases were identified using the ICDD PDF-2 (2017). The unit-cell parameters were calculated using the Le Bail full-profile fitting (the Jana2006 software).

### 2.4. Scanning Electron Microscopy (SEM)

The SEM and mapping/elemental area analysis of the grown crystals were performed on Quanta 200 3D scanning electron microscope (FEI, Hillsboro, IL, USA) equipped with EDX (EDAX, Hillsboro, IL, USA).

### 2.5. Single-Crystal X-ray Diffraction Study of $KR_3F_{10}$

For both compounds, $KTb_3F_{10}$ and $KY_3F_{10}$ suitable crystals were selected and mounted on the Rigaku XtaLAB Synergy-DW diffractometer equipped with HyPix-Arc 150 detector. Single-crystal X-ray diffraction data were collected at room temperature using monochromatized MoK$\alpha$-radiation for $KTb_3F_{10}$ and AgK$\alpha$-radiation for $KY_3F_{10}$. The intensities were corrected for numerical absorption based on Gaussian integration over a multifaceted crystal model using the CrysAlisPro software [45]. The crystal structures of $KR_3F_{10}$ ($R$ = Y, Tb) were solved by the intrinsic phasing method using ShelXT [46] structure solution program with Olex2 [47] and refined with the anisotropic displacement parameters for all sites using ShelxL [48] by the full-matrix least-squares method.

### 2.6. Optical Properties

Transmission spectra of the crystals were recorded under room temperature using a Varian Cary 5000 spectrophotometer (Agilent Technologies, Santa Clara, CA, USA) in the spectral region $\lambda$ = 0.19–3.30 $\mu$m. Samples for investigation were taken from the middle part of the crystal ingots.

### 2.7. The Thermal Conductivity Measurements

The temperature dependence of crystal thermal conductivity $k$(T) was measured by an absolute steady-state axial heat flow technique in the temperature range of 50–300 K. The measurement procedure was described in detail in [49]. The samples represented non-oriented parallelepipeds with an approximate size of $6 \times 6 \times 20$ mm$^3$. The error in determining the absolute $k$ value did not exceed $\pm$5%.

### 2.8. The Electrical Conductivity Measurements

The electrical conductivity $\sigma_{dc}$ of the $KR_3F_{10}$ crystals was determined by impedance spectroscopy. The impedance measurements were carried out in the frequency range of $5$–$5 \times 10^5$ Hz and the resistance range of $1$–$10^7$ Ohm using a Tesla B-507 impedance tester at temperatures of 550–825 K in the vacuum ~1 Pa. $KR_3F_{10}$ single crystals have a perfect cleavage along the crystallographic (111) plane. Taking this into account, samples oriented along the crystallographic [111] direction were prepared. The thickness of the samples was about 1.5 mm and the silver electrode areas were about 20–30 mm$^2$. Silver paste (Leitsilber) was used as a current-conducting electrode. The relative measurement error did not exceed 5%. The presence in the impedance spectra of the blocking effect from inert (silver) electrodes at low frequencies indicates the ionic nature of the electrical transfer.

## 3. Results and Discussion

### 3.1. Growth Process Results and Crystals Characterization

The utilization of KF as a precursor is absolutely unsuitable for the production of oxygen-free $KR_3F_{10}$ crystals (see Figure 3a) as shown by our growth experiments. These crystals appeared cloudy and opalescent or contained cloudy inclusions in the bulk. The use of a hard fluorinating growth atmosphere (fully consisting of $CF_4$) did not lead to positive results.

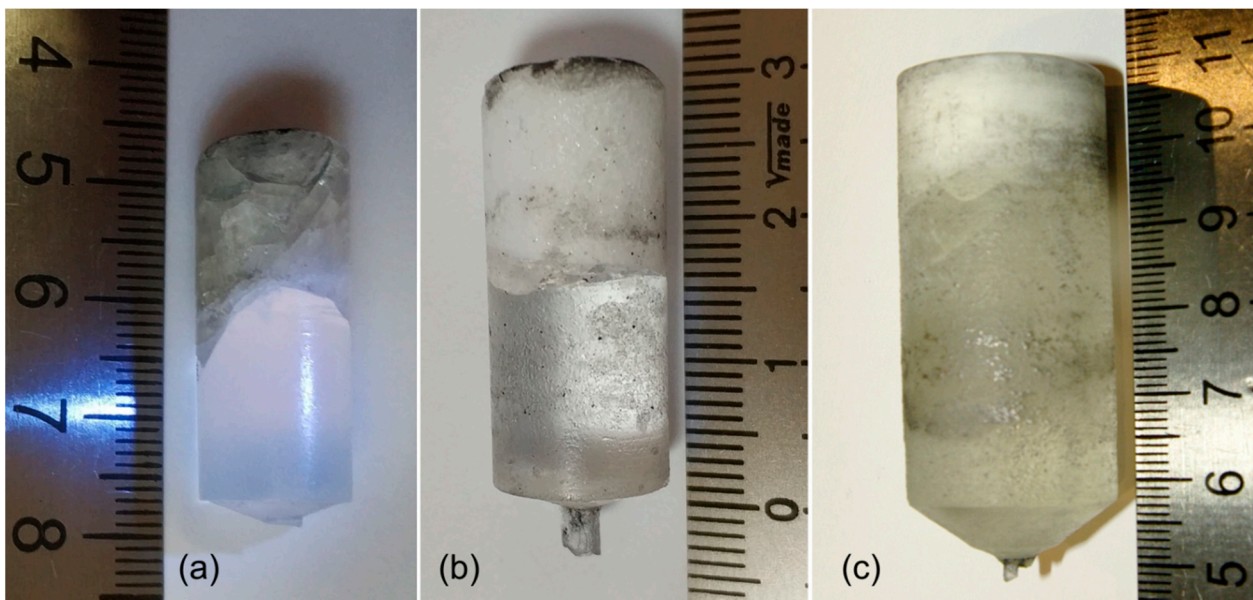

**Figure 3.** Appearance of as-grown $KY_3F_{10}$ (KYF) crystals with traces of oxygen, obtained using a commercial reagent KF (**a**); transparent KYF (**b**) and $KTb_3F_{10}$ (KTF) (**c**) crystals grown using potassium hydrofluoride as an initial reagent.

The use of potassium hydrofluoride in the synthesis of $KR_3F_{10}$ single crystals has shown significant advantages. Hydrofluoride $KHF_2$ decomposes according to the scheme: $KHF_2 = KF + HF$ in the temperature range from 430 to 450 K with the evolution of anhydrous hydrogen fluoride [50], which has an additional fluorinating effect on the melt during growth process. $KR_3F_{10}$ crystals grown using potassium hydrofluoride were colorless and transparent in ambient light, and did not contain scattering inclusions (Figure 3b,c). In some cases, crystals had cracks due to perfect cleavage along the (111) planes if high cooling rates (over 75 K/h) were applied.

The assignment of grown crystals to the $CaF_2$ structure type (space group $Fm\bar{3}m$, Z = 8) was confirmed by XRD. The diffraction patterns of the $KR_3F_{10}$ crystals are shown in Figure 4. Transparent crystal parts are single phase. The cubic lattice parameter of the KYF crystal is $a = 11.5468(1)$ Å at room temperature, which is confirmed by the published data (coincides with the standard patterns PDF #75-3059). The composition of the transparent part of KTF crystal is not constant and represents a partial solid solution; a change in the lattice parameter $a$ from 11.679(1) to 11.663(1) Å is observed along the length of the ingot, respectively. The structural aspects of the formation of such a solid solution and the mechanism of its nonstoichiometry require detailed study in the future. Crystal density $\rho = 4.262(5)$ g/cm$^3$ for KYF (measured by hydrostatic weighing in distilled water) is insignificantly lower than the theoretical density data for KYF. For the terbium analogue, $\rho = 5.806(5)$ g/cm$^3$.

Losses during growth (up to 1 wt. %) are mainly due to the evaporation of more volatile KF. Therefore, a shift of the composition to the $RF_3$-enriched region occurs. As a result of incongruent melting, additional impurity phases were detected in the parts of the $KR_3F_{10}$ crystalline boules, which crystallized last. SEM and XRD revealed the presence of additional impurity of $YF_3$ in the opaque part of the KYF crystal. The eutectic (KYF + $YF_3$) mixture and the precipitated $YF_3$ rod-like phase in the bulk KYF matrix were clearly observed (Figure 5).

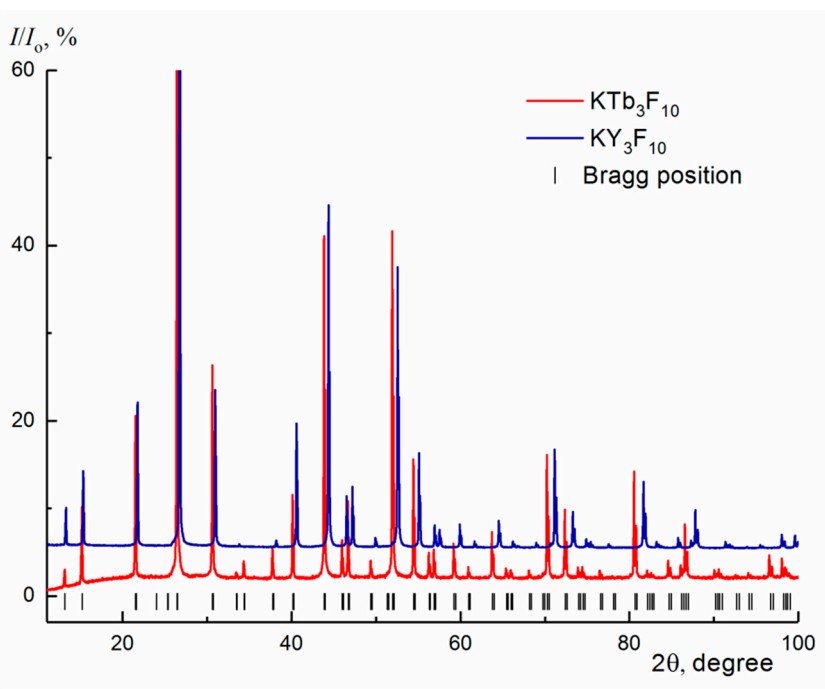

**Figure 4.** XRD patterns of the KYF and KTF crystals. The positions of Bragg reflection peaks for KTF (space group $Fm\bar{3}m$) are indicated.

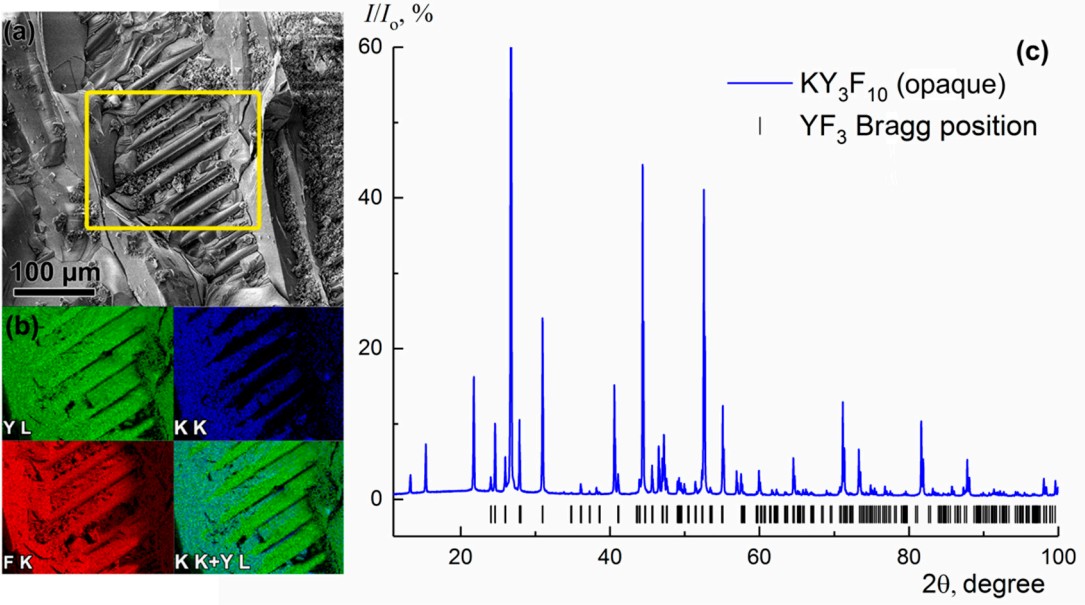

**Figure 5.** SEM image of opaque KYF crystal part (**a**) and corresponding EDX elemental mapping (**b**) of area marked in (**a**): Y(L)—green, K(K)—blue, F(K)—red and overlapped color map (K + Y); XRD patterns of KYF opaque crystal part, illustrating the presence of two phases simultaneously—$KY_3F_{10}$ and $YF_3$ (**c**). The positions of Bragg reflection peaks for $YF_3$ phase (space group *Pnma* with lattice parameters $a = 6.3667(1)$, $b = 6.8583(1)$, $c = 4.3930(1)$ Å) are indicated.

A decrease in the degree of melt overheating during growth results in an increase in the length of the useful transparent part of the KYF crystals. Two polymorphic modifications, $KTb_2F_7$ and the compound $KTbF_4$, were detected in addition to the main cubic phase in the top parts of the grown KTF crystals, which crystallized last, as shown in [25].

### 3.2. Crystal Structure Refinement

The crystallographic parameters of $KR_3F_{10}$, experimental conditions of the data collection, and the structural refinement results are summarized in Table 1. Atomic positions and displacement parameters for $KR_3F_{10}$ compounds, and selected interatomic distances for $KTb_3F_{10}$ and $KY_3F_{10}$ crystals, are given in Tables S1, S2 and S3 respectively. CCDC 2062503 ($KTb_3F_{10}$) and 2062504 ($KY_3F_{10}$) contain the supplementary crystallographic data for this paper. The data can be obtained free of charge from The Cambridge Crystallographic Data Centre via www.ccdc.cam.ac.uk/structures (accessed on 14 March 2021).

**Table 1.** Crystal data for $KTb_3F_{10}$ and $KY_3F_{10}$ structures.

| Chemical Formula | $KTb_3F_{10}$ | $KY_3F_{10}$ |
|---|---|---|
| Crystal system, space group | \multicolumn Cubic, $Fm\overline{3}m$ | |
| $a$ (Å) | 11.67246 (4) | 11.5384 (1) |
| $V$ (Å$^3$) | 1590.33 (2) | 1536.16 (4) |
| $Z$ | 8 | |
| Crystal size (mm) | $0.1 \times 0.1 \times 0.07$ | $0.21 \times 0.2 \times 0.07$ |
| $\mu$ (mm$^{-1}$) | 27.046 | 12.575 |
| Radiation type | Mo $Ka$, $\lambda = 0.71073$ Å | Ag $Ka$, $\lambda = 0.56087$ Å |
| Theta range for data collection (°) | 3.0–37.7 | 2.4–30.7 |
| Limiting indices | $-19 \leq h \leq 19$ $-19 \leq k \leq 19$ $-19 \leq l \leq 20$ | $-20 \leq h \leq 20$ $-20 \leq k \leq 20$ $-19 \leq l \leq 20$ |
| Number of measured, independent and observed [$I > 2\,\sigma(I)$] reflections | 31006, 264, 260 | 27047, 294, 294 |
| Data/restrains/parameters | 264/0/13 | 294/0/13 |
| $R_{int}$ | 0.027 | 0.067 |
| $R[F^2 > \sigma 2(F^2)]$, $wR(F^2)$, $S$ | 0.008, 0.028, 1.01 | 0.0137, 0.033, 1.28 |
| Largest diff. peak and hole (e/Å$^3$) | 0.53 and $-1.10$ | 0.53 and $-0.92$ |
| Absorption correction | Numerical absorption correction based on gaussian integration over a multifaceted crystal model *CrysAlis PRO* 1.171.41.95a | |
| $T_{min}$, $T_{max}$ | 0.196, 0.349 | 0.138, 0.652 |
| Extinction correction: SHELXL2018/3 | $Fc^* = kFc[1 + 0.001 \times Fc^2 l^3/\sin(2q)]^{-1/4}$ | |
| Extinction coefficient | 0.000255(18) | 0.0128(4) |

Computer programs: *CrysAlis PRO* 1.171.41.95a [45], SHELXT 2018/2 [46], Olex2 1.3 [47], *SHELXL* 2018/3 [48].

$KTb_3F_{10}$ and $KY_3F_{10}$ compounds are isostructural and represent the structure type $Zr_3PbO_4F_6$ [51]. Our crystal structure solution results confirm previous structural studies of $KTb_3F_{10}$ [20,52] and $KY_3F_{10}$ [27,28,53]. The asymmetric unit of $KR_3F_{10}$ contains one rare-earth cation $R^{3+}$ (24e: 0.24, 0, 0), one anion $K^+$ (8c: $\frac{1}{4}$, $\frac{1}{4}$, $\frac{1}{4}$), and two distinct sites of fluorine ions: F(1) (48i: 1/2, 0.1667, 0.1667) and F(2) (32f: 0.112, 0.112, 0.112). Rare-earth cations $R^{3+}$ are bonded to eight fluorine ions forming square antiprisms. The crystal structure of $KR_3F_{10}$ can be described by considering two types of fundamental building blocks—polyanionic clusters of six $RF_8$ square antiprisms connected either along edges—$[R_6F_{32}]$ cluster or along vertices—$[R_6F_{36}]$ ones [20,27,28,52–55].

Taking into account the shortest distances between the $R^{3+}$ cations, one can consider the $[R_6F_{32}]$ cluster, in which the 24 external F(1) fluorine atoms form a truncated octahedron, and eight internal F(2) atoms form a cubic central cavity (Figure 6, left). The alternative means of describing the crystal structure is based on the polyanionic cluster $[R_6F_{36}]$ in which 12 internal F(1) atoms form a central cuboctahedral cavity and 24 outer F(2) atoms form a rhombicuboctahedron (Figure 7, left). In both cases, the clusters and $K^+$ ions are arranged as $Ca^{2+}$ and $F^-$ ions in the fluorite structure (Figures 6 and 7). The clusters $[R_6F_{36}]$ or $[R_6F_{32}]$ adopt a face-cubic centered arrangement, occupying the vertices and face centers

of the cubic unit cell, and are connected through F–F edges to form a three-dimensional framework. The $K^+$ ions occupy the tetrahedral interstitial sites in fluorite-like structure and are coordinated with four nearest fluorine ions F(2) at distances of 2.786 Å in $KTb_3F_{10}$ or 2.766 Å in $KY_3F_{10}$, and to twelve fluorine atoms F(1) at a distance of 3.226 Å in $KTb_3F_{10}$ or 3.195 Å in $KY_3F_{10}$ (Figure 7, right).

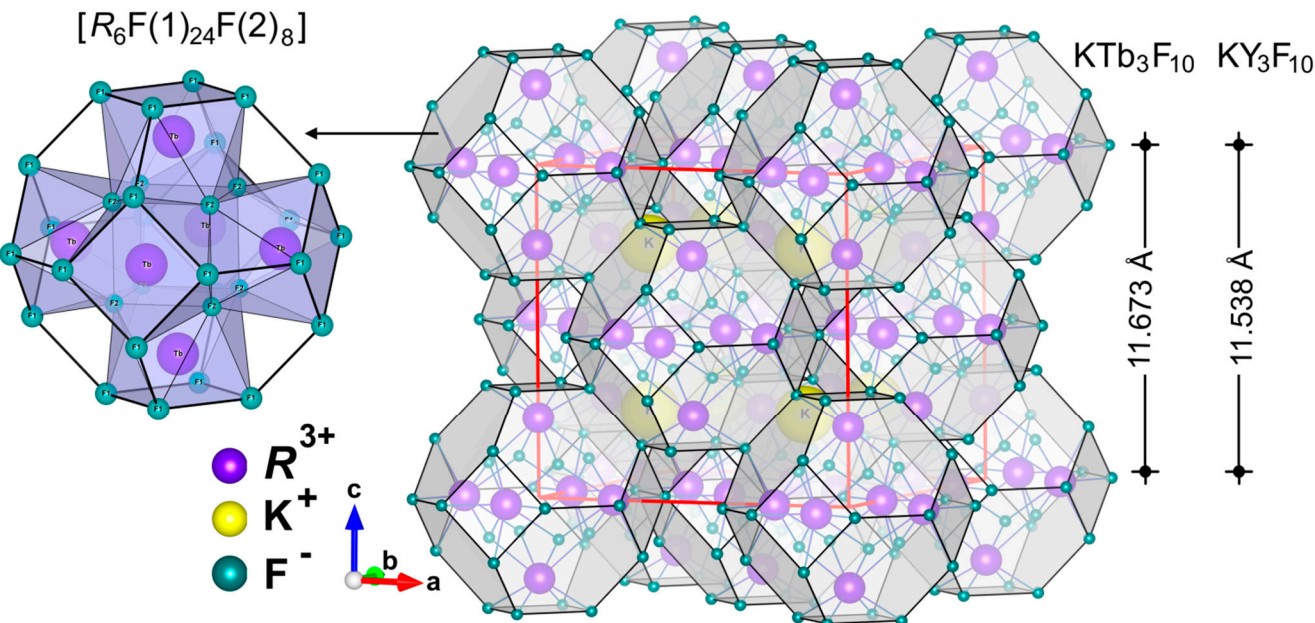

**Figure 6.** Geometry of the polyanionic cluster $[R_6F_{32}]$ (**left**) and crystal structure of $KR_3F_{10}$ ($R$ = $Tb^{3+}$, $Y^{3+}$) as a three-dimensional assembly of $[R_6F_{32}]$ clusters (**right**).

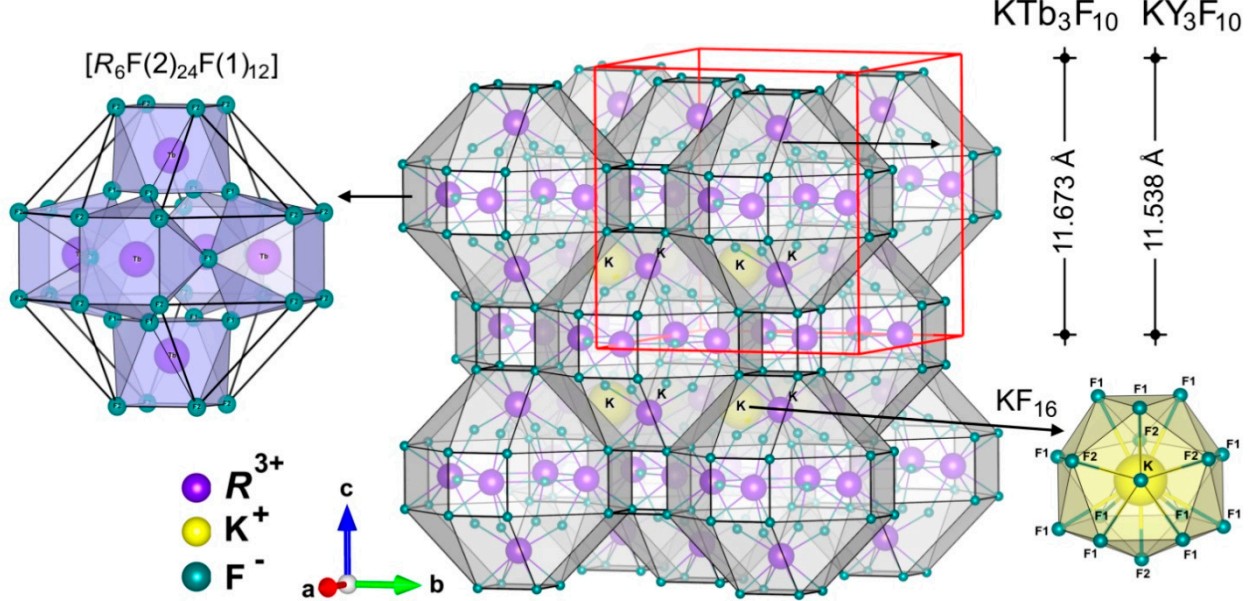

**Figure 7.** Geometry of the polyanionic cluster $[R_6F_{36}]$ (**left**), the crystal structure of $KR_3F_{10}$ ($R$ = $Tb^{3+}$, $Y^{3+}$) as a three-dimensional assembly of $[R_6F_{36}]$ clusters (**center**), and $KF_{16}$-polyhedron (**right**).

The composition of the polyanionic $[R_6F_{36}]$ cluster is similar to that of the rare-earth cluster in the ordered phases of the $MF_2$–$RF_3$ systems (where *M*—alkaline earth, *R*—rare-

earth cations). This reveals the affinity of the $KR_3F_{10}$ structural type with other fluorite-like ordered phases containing anionic $[R_6F_{36}]$ cuboctahedra. This approach makes it possible to consider $KR_3F_{10}$ crystals as representatives of fluorite-like phases $MF_2$–$RF_3$ and to describe their structurally dependent properties from the point of view of general structural nature [56].

### 3.3. Optical Properties of $KR_3F_{10}$ Crystals

The optical absorption spectra of the grown $KR_3F_{10}$ crystals are shown in Figure 8a, b. Both transparent and opalescent KYF crystals exhibit high absorption in the ultraviolet spectral region (Figure 8a). Significant increase in the overall absorption level in the visible region due to strong light scattering was observed for the opalescent oxygen-content KYF samples. Their short-wavelength transmission limit was significantly shifted to the visible spectral region.

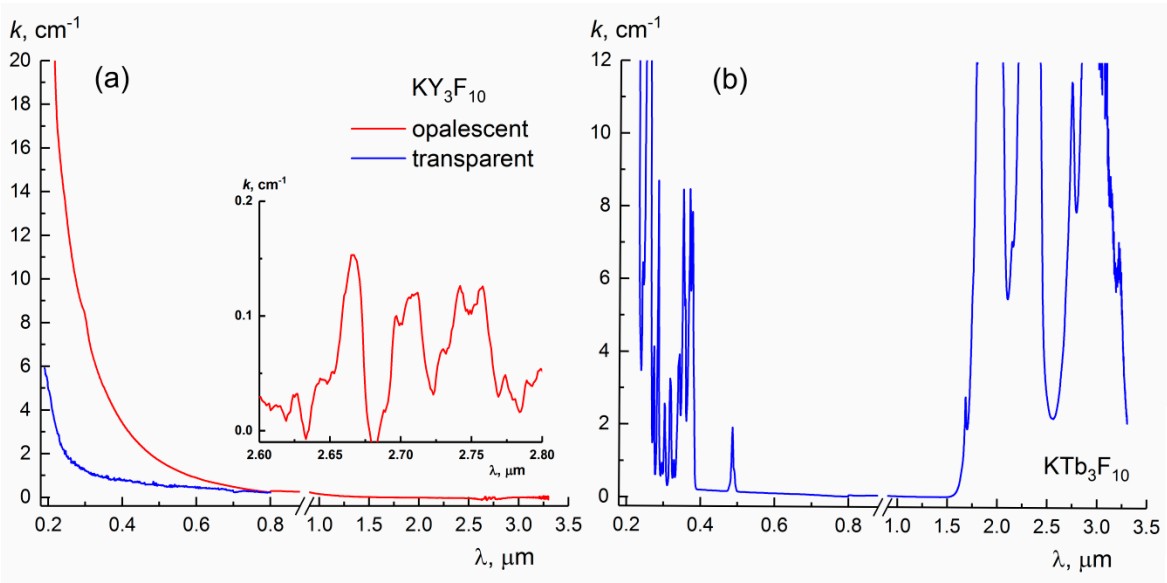

**Figure 8.** Absorption spectra of the transparent and opalescent samples of $KY_3F_{10}$ (**a**) and $KTb_3F_{10}$ single crystals (**b**). Insert: enlarged part of spectrum for KYF crystal in infrared region.

A number of additional weak absorption bands were observed in the IR range for the opalescent KYF crystal (Figure 8a, inset). It is known that some oxygen impurities, namely $OH^-$ groups and $HCO^-$ complexes, are responsible for the appearance of absorption bands in the range $\lambda$ = 2.6–3.5 $\mu$m [57]. As seen from Figure 8a, the IR spectrum contains weakly intense narrow bands that can be attributed to these complexes. The $OH^-$ and $HCO^-$ complexes are due to carbon contamination caused by oxides, which is not totally eliminated in the synthesis process and indicates that there were water vapor traces during the crystal growth process. No IR absorption bands appeared in this spectral range for the transparent KYF samples. Additional deep purification of the initial charge and atmosphere during the growth process can significantly improve the spectral quality of these crystals in the short-wavelength part of the transparency window. The last is highly sensitive to the oxygen impurity contaminations. Note that oxygen-free KYF crystals are characterized by a wide transparency region up to 0.13 $\mu$m and are promising optical materials for the VUV spectral region [58,59].

The absorption spectrum of $KTb_3F_{10}$ crystal is represented in Figure 8b. This is typical for crystals containing $Tb^{3+}$ ions. The electro-dipole transitions within the $4f^8$ configuration of this ion are clearly observed without additional impurity lines [18,20,21]. $KTb_3F_{10}$ crystals demonstrate a transparency window in the range of 0.4–1.5 $\mu$m, with the exception of a narrow absorption line near $\lambda$ = ~485 nm ($7F^6$–$5D^4$ transition of $Tb^{3+}$ ion).

### 3.4. Thermal Conductivity Measurements

The thermal conductivity of $KTb_3F_{10}$ crystals was measured for the first time in a wide temperature range (Figure 9a). The thermal conductivity temperature dependences $k(T)$ of KYF [60], $Na_{0.4}Y_{0.6}F_{2.2}$, and $Na_{0.37}Tb_{0.63}F_{2.26}$ [61] crystals are shown for comparison (Figure 9b). It is noticeable that the $k(T)$ dependences of the $KR_3F_{10}$ and $Na_{0.5-x}R_{0.5+x}F_{2+2x}$ crystal families differ significantly. The presence of a large number of phonon scattering centers in $Na_{0.5-x}R_{0.5+x}F_{2+2x}$ crystals determines the glass-like character of their $k(T)$ dependences.

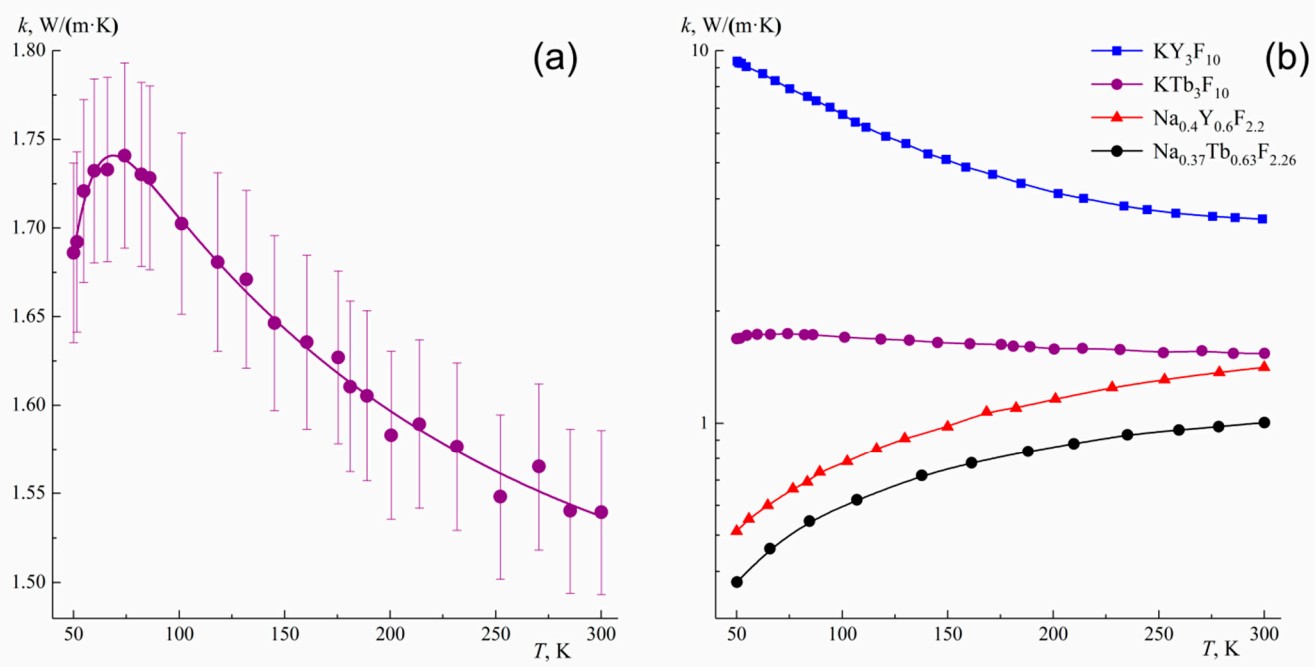

**Figure 9.** Temperature dependence of the thermal conductivity $k(T)$ of $KTb_3F_{10}$ crystal (**a**) and comparative data on the $k(T)$ for KF- and NaF-based crystals (**b**). The vertical frames correspond to the reproducibility limits of experimental results within ±3% for the possibility of the tracking temperature trend.

The thermal conductivity of the $KTb_3F_{10}$ crystal is also low; it varies within narrow limits, from 1.54 to 1.74 $Wm^{-1}K^{-1}$ in the explored temperature range. Amorphous materials have similar values of the thermal conductivity coefficient. However, in the region of $T$ = 74 K, a low-temperature maximum $k(T)$ characteristic of crystalline media was observed for $KTb_3F_{10}$. These features indicate, on the one hand, the presence of a long-range order in the crystal structure of this material, and, on the other hand, a very significant phonon scattering manifestation in the investigated temperature range.

The KTF crystal is significantly inferior to its yttrium isostructural KYF analogue in terms of thermal conductivity. For comparison, the composition of KTF contains $K^+$ and $Tb^{3+}$ cations, which differ greatly in mass, whereas the corresponding difference is less in KYF crystal. A significant difference in the masses of the oscillators predetermines the presence of optical modes in the phonon spectrum of a crystal, which usually make a small contribution to heat transfer compared to acoustic modes. So this factor makes the crystal a poor heat conductor. In addition, a higher density of the $KTb_3F_{10}$ crystal corresponds to a lower average propagation velocity of acoustic vibration modes.

Another possible factor determining the comparatively low thermal conductivity of $KTb_3F_{10}$ crystal should be indicated. In many cases, $Tb^{3+}$ ions exhibit splitting of the electron paramagnetic levels of the $4f$ shell. Oxide Tb-based crystals (differing from fluoride crystals due to a stronger crystal field) are characterized by a relatively low thermal conductivity [62]. If the magnitude of the splitting $\Delta E$ is in the range of 0–200 $cm^{-1}$, then it

should be the cause of resonant phonon scattering and a corresponding decrease in thermal conductivity. Unfortunately, no data on this level splitting in the $KTb_3F_{10}$ can be found in the literature.

Disorder in the crystal structure usually results in phonon scattering. The basic fluorite structure of $KR_3F_{10}$ crystals is characterized by a divalent oxidation state of cations. The presence of trivalent rare-earth ions causes the appearance of large defect clusters, which are effective centers of phonon scattering. The consequence of this is a significant decrease in thermal conductivity and a weakening of its $k(T)$ dependence. This phenomenon is well known (see, for example, [63–65]) for the heterovalent solid solution $M_{1-x}R_xF_{2+x}$ ($M$ = Ca, Sr, Ba; $R$—rare-earth) crystals with a fluorite structure. The behavior of the thermal conductivity of $M_{1-x}R_xF_{2+x}$ crystals becomes characteristic of glasses due to percolation of clusters at high concentrations of trivalent ions. Apparently, the presence of $[R_6F_{36}]$ structural blocks and their ordering can determine the thermal conductivity in the case of $KR_3F_{10}$ crystals. Therefore, the temperature dependence of the thermal conductivity of $KR_3F_{10}$ crystals can be considered as transitional from $k(T)$ of undoped $MF_2$ crystals to $k(T)$ of concentrated heterovalent $M_{1-x}R_xF_{2+x}$ solid solutions.

The results of a comparison of the thermal conductivity of $Na_{0.4}Y_{0.6}F_{2.2}$ and $Na_{0.37}Tb_{0.63}F_{2.26}$ crystals were as expected (Figure 9b). The presence of $Tb^{3+}$ ions in the crystal composition in this case is also a negative factor and leads to its significant decrease, converting crystals of this type into low-temperature heat insulators.

### 3.5. Ionic Conductivity Measurements

The temperature dependences of ionic conductivity $\sigma_{dc}(T)$ for different crystal samples in the $KF$–$YF_3$ system are shown in Figure 10a. It can be seen that the transparent and opalescent KYF crystal fragments have the same $\sigma_{dc}$ values. The conductivity for the eutectic ($KY_3F_{10}$ + $YF_3$) composite is higher than for the pure KYF compound. The extrapolated $\sigma_{dc}$ values at 500 K are $1.7 \times 10^{-9}$ and $1.2 \times 10^{-10}$ S/cm for the composite and KYF single crystal, respectively. The comparison $\sigma_{dc}(T)$ dependences for $KR_3F_{10}$ crystals are shown in Figure 10b. Data for $Na_{0.5-x}R_{0.5+x}F_{2+2x}$ solid solution crystals are shown additionally.

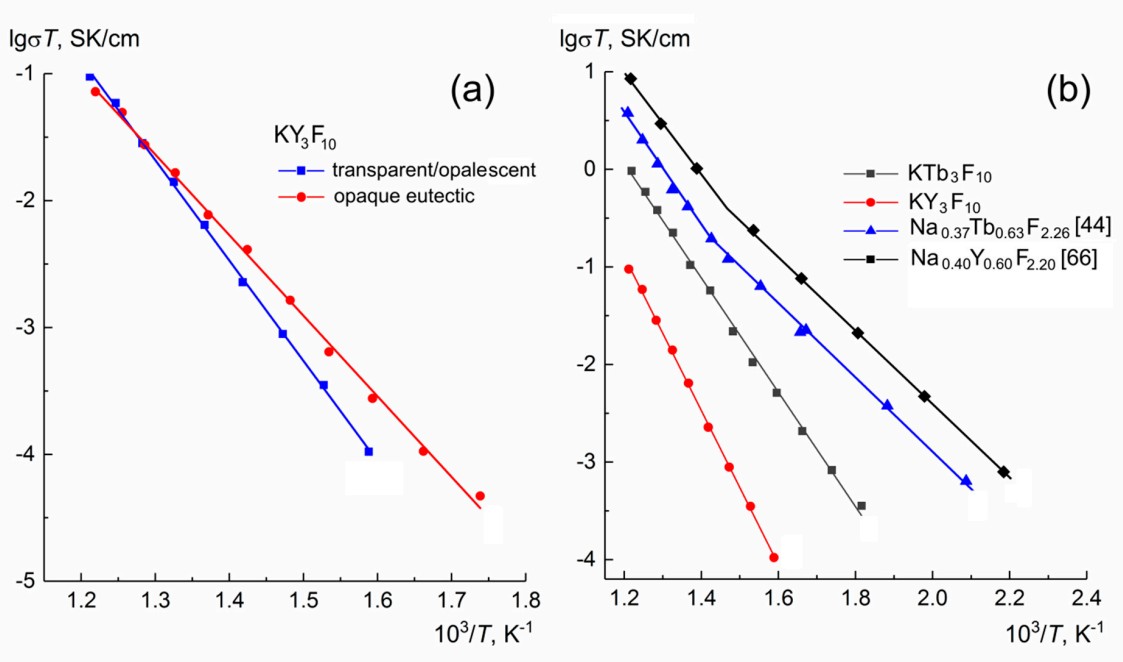

**Figure 10.** Temperature dependences of ionic conductivity for the samples from different $KY_3F_{10}$ crystal fragments (**a**); comparative ionic conductivity data for KF- and NaF-based single crystals (**b**).

The $\sigma_{dc}(T)$ dependences for $KR_3F_{10}$ ($R$ = Tb, Y) single crystals were fitted according to the Arrhenius–Frenkel equation:

$$\sigma_{dc}T = A\exp(-H_\sigma/k_BT),$$

where $A$—preexponential conductivity factor, $H_\sigma$—activation enthalpy of ion transport, $k_B$—Boltzmann's constant, $T$—temperature. The Arrhenius–Frenkel equations parameters and measured room temperature thermal conductivity data are given in Table 2. The ionic conductivity of KYF was much higher than that of KTF, because KYF has a lower potential barrier for the charge carrier migration ($H_\sigma$) than that of KTF.

**Table 2.** Physical properties of $KR_3F_{10}$ and $Na_{0.5-x}R_{0.5+x}F_{2+2x}$ ($R$ = Tb, Y) crystals.

| Parameter | $KTb_3F_{10}$ | $KY_3F_{10}$ | $Na_{0.37}Tb_{0.63}F_{2.26}$ | $Na_{0.4}Y_{0.6}F_{2.2}$ |
|---|---|---|---|---|
| $A$, (SK)/cm | $1.2 \times 10^7$ | $4.0 \times 10^8$ | $3.6 \times 10^4$ [44] | $3.1 \times 10^4$ [66] |
| $H_\sigma$, eV | 1.16 (550–820 K) | 1.57 (630–825 K) | 0.74 (478–700 K) | 0.80 (380–730 K) |
| $\sigma_{dc}$(500 K), S/cm | $4.9 \times 10^{-8}$ | $1.2 \times 10^{-10}$ | $2.6 \times 10^{-6}$ | $4.8 \times 10^{-6}$ |
| $k$(300 K), W/(mK) | 1.7 1.67 [67] | 3.5 [60] | 1.0 [61] | 1.4 |

The $\sigma_{dc}$ value for $KTb_3F_{10}$ is two orders of magnitude higher than for $KY_3F_{10}$ crystals. Nevertheless, results obtained for $KR_3F_{10}$ compounds and the electrophysical data for the $Na_{0.5-x}R_{0.5+x}F_{2+2x}$ solid solutions show that the fluorine-ion transfer in the KF-based compounds is significantly lower than in the NaF-based ones (Figure 10b). This drop in ionic conductivity magnitude is primarily associated with a twofold increase in potential barriers to the migration of charge carriers: from 0.7–0.8 eV for $Na_{0.5-x}R_{0.5+x}F_{2+2x}$ ($R$ = Tb, Y) to 1.2–1.6 eV for $KR_3F_{10}$ crystals. The $\sigma_{dc}(T)$ dependences for $Na_{0.5-x}R_{0.5+x}F_{2+2x}$ solid solution crystals consist of two linear segments with the temperature of the transition between them of T = ~750 K (Figure 10b). Each segment of conduction dependences satisfies the Arrhenius–Frenkel equation. A similar situation is valid for fluorite-type $Ca_{1-x}R_xF_{2+x}$ ($R$ = La–Lu, Y) crystals [68]. A common feature of the conductivity of these crystals is the fulfillment of the following condition: activation enthalpy of ion transport in the high-temperature region is greater than in the low-temperature region. The high-temperature region of conductivity is probably connected with the process of dissociation of bonded fluorine ions from clusters.

According to structural studies of disordered $Na_{0.5-x}R_{0.5+x}F_{2+2x}$ (R = Y, Ho, Yb) solid solutions [69–71], the octahedral clusters $[R_6F_{37}]$ are formed in their crystalline lattices. Taking into account the nearest cationic ($M$) environment of $[R_6F_{37}]$ clusters, they can be represented as superclusters $\{M_8[R_6F_{37}]F_{32}\}$, where $8M^{2+} = 4Na^+ + 4R^{3+}$ [56]. The $\{M_8[R_6F_{37}]F_{32}\}$ clusters at $M^{2+} = Ca^{2+}$, $Sr^{2+}$, $Ba^{2+}$ are also formed in nonstoichiometric fluorites of $Ca_{1-x}R_xF_{2+x}$ (R = Dy–Lu, Y), $Sr_{1-x}R_xF_{2+x}$ (R = Nd–Lu, Y) and $Ba_{1-x}R_xF_{2+x}$ ($R$ = La–Lu, Y) and $Ca_2YbF_7$, $Sr_4Lu_3F_{17}$, $Ba_4R_3F_{17}$ (for $R$ = Yb, Y) ordered ones [68,72–74]. Ionic transfer in $Na_{0.5-x}R_{0.5+x}F_{2+2x}$ solid solutions occurs in the anionic sublattice and is caused by the hopping migration of $F^-$ ions according to the interstitial mechanism. The scheme of heterovalent substitutions in the fluorite structure of $Na_{0.5-x}R_{0.5+x}F_{2+2x}$ has the following form (block isomorphism model) [75]:

$$\{(Na,R)_{14}F_{64}\}^{36-} \rightarrow \{(Na_{0.5}R_{0.5})_8[R_6F_{37}]F_{32}\}^{35-} + F_{mob}{}^-,$$

where $F_{mob}{}^-$ is a mobile charge carrier.

Thus, the rare-earth sublattice in nonstoichiometric fluorite-type $Na_{0.5-x}R_{0.5+x}F_{2+2x}$ phases is disordered and only short-range order (octahedral clusters) is observed. In the superstructural fluorite-type $KR_3F_{10}$ ($K_{0.25}R_{0.75}F_{2.50}$ composition) phases, long-range order appears [28,52,56] in the cationic and anionic sublattices. Structural ordering in $KR_3F_{10}$

crystals leads to an increase in potential barriers to carrier migration and a decrease in conductivity by 2–4 orders of magnitude.

## 4. Conclusions

Bulk $KR_3F_{10}$ ($R$ = Y, Tb) single crystals were successfully grown by the Bridgman technique. The synthesis, growth parameters, and investigation of properties are presented for crystals of this type in detail.

Thermal conductivity $k(T)$ of $KR_3F_{10}$ crystals experimentally determined for the first time in the temperature range of 50–300 K noticeably exceeds the thermal conductivity of isostructural $Na_{0.5-x}R_{0.5+x}F_{2+2x}$ solid solution crystals ($R$ = Tb, Y). It is shown that the Tb-based fluoride crystals are inferior in thermal conductivity to yttrium analogs due to significant phonon scattering manifestation. The temperature dependences of the ionic conductivity $\sigma_{dc}(T)$ of $KR_3F_{10}$ ($R$ = Y, Tb) crystals were investigated. $\sigma_{dc}(T)$ dependences satisfy the Frenkel–Arrhenius equation in the temperature range 550–820 K. Comparison of the electroconductive properties of $KR_3F_{10}$ compounds and $Na_{0.5-x}R_{0.5+x}F_{2+2x}$ solid solutions, the structure of which builds on the $[R_6F_{37}]$ clusters basis, is performed. The ordered $KR_3F_{10}$ superstructure formed on the basis of these clusters leads to a twofold increase in potential barriers for the migration of charge carriers and a drop in ionic conductivity (at 500 K) by 2–4 orders of magnitude in $KR_3F_{10}$ single crystals compared to $Na_{0.5-x}R_{0.5+x}F_{2+2x}$ solid solutions with a disordered cluster formation.

These complex studies, from crystal growth to structure determination and study of properties, will be of great importance for photonic applications of complex fluorides $KR_3F_{10}$ ($R$ = Y, Tb) in the future.

**Supplementary Materials:** The following are available online at https://www.mdpi.com/2073-4352/11/3/285/s1, Table S1: Atomic positions and displacement parameters for $KR_3F_{10}$ ($R$ = Y, Tb) crystals, Table S2: Selected interatomic distances for $KTb_3F_{10}$ single crystal, Table S3: Selected interatomic distances for $KY_3F_{10}$ single crystal, CIF file for $KTb_3F_{10}$ single crystal, CIF file for $KY_3F_{10}$ single crystal.

**Author Contributions:** D.N.K., I.I.B., N.A.A., P.A.P., N.I.S. performed the experiments, prepared figures and manuscript; D.N.K., I.I.B. performed crystal growth experiments; N.A.A. and P.A.P. analyzed crystal quality; N.I.S. performed conductivity investigation; N.A.A., A.G.I. performed crystal structure analysis; A.G.S. performed spectroscopic investigation; P.A.P. performed thermal conductivity measurements; D.N.K., I.I.B., A.G.I., P.A.P. analyzed the data, interpreted experiments; D.N.K. provided the idea, designed the experiments; D.N.K. coordinated the scientific group. All authors have read and agreed to the published version of the manuscript.

**Funding:** This research was funded by the Russian Foundation for Basic Research (project 19-02-00877) in the part concerning growth of the crystals and by the Ministry of Higher Education and Science of the Russian Federation within the State assignments of the Federal Scientific Research Centre "Crystallography and Photonics" of the Russian Academy of Sciences in the part concerning investigation and analysis of crystal properties using the equipment of the Shared Research Center (project RFMEFI62119X0035).

**Conflicts of Interest:** The authors declare no conflict of interest. The funders had no role in the design of the study; in the collection, analyses, or interpretation of data; in the writing of the manuscript, or in the decision to publish the results.

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
