# Peer review of "Growth Peculiarities and Properties of KR3F10 (R = Y, Tb) Single Crystals"

_crystals, doi:10.3390/cryst11030285_

Round 1

Reviewer 1 Report

I find this paper to include some interesting and useful new results on compounds of interest to both academic and industry, for their physical properties. The new methodology of crystal growth, in addition to the reporting of thermal and ionic conductivity data certainly make the paper suitable for publication in this journal. I have only relatively minor comments:

  1. The single crystal XRD data (Tables 2-4) could be deposited as Supplementary material outside the main paper: both crystal structures have been reported previously, although the quality of the present study does appear to be better than the previous ones.
  2. line 206 XRD should be XRD
  3. line 210 - clarify what exactly the 'limited solid solution' is, in terms of the range of chemical composition. Is this a range of cation stoichiometry, or related to oxygen inclusion? How is this accommodated in the crystal structure?
  4. Fig 5c 'Bregg' should be 'Bragg'.
  5. Overall standard of written English is very good, though there are a few missing articles ('the', 'a', etc) which should be corrected

Author Response

Response to Reviewer Comments

Reviewer #1

We would like to express our gratitude to the reviewer for checking the manuscript and making minor comments that will help make it clearer for readers and more perfect.

Comment 1

The single crystal XRD data (Tables 2-4) could be deposited as Supplementary material outside the main paper: both crystal structures have been reported previously, although the quality of the present study does appear to be better than the previous ones.

Response 1

Thank you for this remark.

The data of Tables 2-4 have been transferred to the Supplementary Materials section.

Comment 2

line 206 XRD should be XRD

Response 2

We corrected this unfortunate typo.

Comment 3

line 210 - clarify what exactly the 'limited solid solution' is, in terms of the range of chemical composition. Is this a range of cation stoichiometry, or related to oxygen inclusion? How is this accommodated in the crystal structure?

Response 3

This question is key for this type of crystals and you are looking at the root of the problem!

n this work, by the term «limited solid solution» we mean only the presence of some phase inhomogeneous in composition. The study of nonstoichiometry of KR3F10 is a separate problem. It turned out that it cannot be solved immediately. Structural studies of KTb3F10 did not reveal the nonstoichiometry by the cations. Studies of properties such as density and refractive index did not provide unambiguous information either. We put our hope in the chemical analysis of the composition. But this will be the next step in our research for the KR3F10 family and it will probably solve this problem.

At present, solid solutions based on difluorides with a fluorite structure M1-xRxF2+x (in the space group Fm-3m) are being studied in detail. Hundreds of works have been devoted to this problem, and dozens of models of defect clustering have been proposed. In this work, nonstoichiometry was revealed in another type of fluoride fluorite-like phases with a doubled lattice parameter. The fact that this is a different mechanism is unambiguous. And how to correctly describe these crystals: the formula K1+xTb3-xF10-2x or K1+3xTb3-xF10 remains to be clarified in the future.

Comment 4

Fig 5c 'Bregg' should be 'Bragg'

Response 4

We corrected this unfortunate typo.

Comment 5

Overall standard of written English is very good, though there are a few missing articles ('the', 'a', etc) which should be corrected

Response 5

We have corrected syntax and spelling, if possible.

Reviewer 2 Report

The manuscript describes the growth and the structural, optical, thermal and electrophysical characterization of KY3F10 and KTb3F10  single crystals. These are difficult compounds to grow and the type of characterization performed gives important physical information on the compounds that is not easy to find in the literature. For this reason, the topic is very interesting and worth of being published in Crystals.
   Moreover, the paper is well organized and well written. I only have a few issues: 
   1) Fig 8: Comparison of the various curves is not strightforward in %Tr scale because samples have different thicknesses. For example, transmission of sample 2 in NIR is the highest, although the sample is cloudy, but this is also the thinnest sample used. I suggest to convert data to absorption to have a direct comparison between the curves. If scales are too different one trace can be expanded by a certain factor that can be written near the trace.
   2) Fig. 9a: judging from the trend of the data, experimental errors could be overestimated, please check. Moreover, I would rather present Fig. 9b as the first plot and Fig.9a as the expansion of data contained in the other figure for example as an inset.
   3) Fig. 10: please include the legend in the plots, as in Fig. 9b
   4) Fig. 10b: fits for traces 3 and 4 are broken lines: they cannot be the result from the fit with a single Arrhenius-Frenkel equation, please explain.

   The paper can be published after this issues are corrected.

Author Response

Reviewer #2

We are grateful to the reviewer for his appreciation of our manuscript

Comment 1

Fig 8: Comparison of the various curves is not strightforward in %Tr scale because samples have different thicknesses. For example, transmission of sample 2 in NIR is the highest, although the sample is cloudy, but this is also the thinnest sample used. I suggest to convert data to absorption to have a direct comparison between the curves. If scales are too different one trace can be expanded by a certain factor that can be written near the trace.

Response 1

Thank you for your comment. Changes have been made to Figure 8 to make it clearer.

Comment 2

Fig. 9a: judging from the trend of the data, experimental errors could be overestimated, please check. Moreover, I would rather present Fig. 9b as the first plot and Fig.9a as the expansion of data contained in the other figure for example as an inset.

Response 2

Data on the thermal conductivity of KTb3F10 crystals were investigated for the first time and are of great practical interest. We would prefer to keep it in the first place and separate as far as possible. We marked the limits with vertical frames reproducibility of experimental results ± 3%.

We have made a change to the caption for the figure: "The vertical frames correspond to the reproducibility limits of experimental results ± 3%, which determine the possibility of comparing samples and tracking temperature trends".

Comment 3

Fig. 10: please include the legend in the plots, as in Fig. 9b

Response 3

We added a legend in the plots of Fig. 10 by analogy

Comment 4

Fig. 10b: fits for traces 3 and 4 are broken lines: they cannot be the result from the fit with a single Arrhenius-Frenkel equation, please explain

Response 4

The temperature sdc(T)-dependences for crystals Na0.5–xR0.5+xF2+2x (R = Tb, Y) consist of two linear segments with the temperature of the transition between them T ~750 K (blue and black curves in Fig.

10b). Each segment of conduction dependences satisfies Arrhenius-Frenkel equation. A similar situation is valid for fluorite-type Ca1-xRxF2+x (R = La-Lu, Y) crystals. [71]. The specific features common for crystals of Na0.5–xR0.5+xF2+2x and Ca1-xRxF2+x x is the fulfillment of the condition Hs2 > Hs1 , where subscript “1” and “2” refers to a low-temperature region and a high-temperature region respectively. The high-temperature region of conduction is probably connected with the process of dissociation of bonded fluorine ions from clusters.

Corresponding explanation have been added to the text of the manuscript.

Reviewer 3 Report

In the article "Growth peculiarities and properties of KR3F10 (R = Y, Tb) single crystals", the authors describe crystal growth, crystallographic structure, optical and thermal properties of the two compounds KYF and KTF, which are emerging and promising compounds for solid-state lasers and optical isolators, respectively. Given the incomplete literature data on both growth and characterization of KYF and KTF, the study is very interesting for the community. Furthermore, the results are encouraging, and the report is detailed enough to provide new and valuable insights into the growth challenges and the resulting structure-property relationship of these materials. The paper is very well written, detailed where necessary but with concise language and meaningful figures. I have only a few minor questions and recommendations to improve the manuscript, and after that, it should be published in Crystals.
Minor questions and recommendations:
- the KF-TbF3 phase diagram (Fig. 2b) is really different to that known in previous literature, e.g. [22]. Unfortunately, Ref. [25] is not (yet) retrievable; the DOI link is not working. Please check this...
- could you estimate the melting points for KYF and KTF, and which one is higher?
- in line 131, "mol" is still written in Cyrillic. Same for "i" [for "and"] in line 398.
- in line 231 (and again in Fig. 10 and the associated text), you refer to the crystal top ("in the top parts of the grown KTF crystals")? As is Fig. 10 you state that these parts are opaque, I suppose "crystal top" should refer to the intransparent whitish areas of Fig.s 3b and 3c, which are most remote from the seed. OPther reserachers refer to the cone and near-seed region as the "crystal top"... It would be better to describe the parts of the boule with the distance from the seed or the progress of growth (crystallized first and later).
- Are the lower parts of the crystals in Fig. 3b and 3c (KYF+YF eutectic?) just a surface effect or is it found across the volume?
- Fig. 4, insert: "Bragg", not "Bregg"
- line 222, "germinated" is not used in crystal growth, use "precipitated" or "segregated" instead
- in section 3.3, I encourage the authors to discuss the impact of structural defects on the thermal conductivity. While there might be of course a dependence on the crystallographic structure and phonon frequency, the "glass-like" behavior of Na-R-F compounds cannot be explained just with such considerations. Could you estimate the temperature-dependent behavior with simple equations for defect and phonon scattering to see whether the KYF (with its maximum obviously lower than the one for KTF) should have a lower defect scattering or not? The strong difference between KTF and KYF is not really explained by the different masses of Tb and Y in respect to K, except if vacancies were present.
- why is the ionic conductivity of KYF so much higher than the one for KTF? Again, could vacancies play a role here?
- what could be the change in ionic transport observed in the Na-R-F- compounds above 750 K? 

Author Response

Reviewer #3

We are grateful for the reviewer's comments

Comment 1

- the KF-TbF3 phase diagram (Fig. 2b) is really different to that known in previous literature, e.g. [22]. Unfortunately, Ref. [25] is not (yet) retrievable; the DOI link is not working. Please check this...

Response 1

Unfortunately for us this article (ref. [25]) has status "in press" status and will not be available until April 2021. The DOI has not been activated yet.

Comment 2

- could you estimate the melting points for KYF and KTF, and which one is higher?
Response 2

These compounds have similar melting points. KY3F1O crystal melts congruently at 1020°C [ref. 29]. For terbium-based crystals the peritectic temperature is ~ 1018±10 °C [Khaidukov, N.M.; Filatova, T.G.; Ikrami, M.B.; Fedorov, P.P. Morphotropy in lanthanide fluoride series. Inorg. Mater. 1993, 29, 1152–1156]

Comment 3

- in line 131, "mol" is still written in Cyrillic. Same for "i" [for "and"] in line 398.
Response 3

Thank you for this remark. We have made the technical error, and now corrected it in the manuscript.

Comment 4

- in line 231 (and again in Fig. 10 and the associated text), you refer to the crystal top ("in the top parts of the grown KTF crystals")? As is Fig. 10 you state that these parts are opaque, I suppose "crystal top" should refer to the intransparent whitish areas of Fig.s 3b and 3c, which are most remote from the seed. Other researchers refer to the cone and near-seed region as the "crystal top"... It would be better to describe the parts of the boule with the distance from the seed or the progress of growth (crystallized first and later).

Response 4

Thanks for this comment. The discrepancies in the description of different parts of the crystal boule were eliminated in the text of the manuscript. Indeed, the top of the crystal is the part farthest from the seed and crystallized last.

Comment 4

- Are the lower parts of the crystals in Fig. 3b and 3c (KYF+YF eutectic?) just a surface effect or is it found across the volume?

Response 4

The observed opaque part of the crystals (growth direction from the bottom to the top of the figure) in Fig. 3a and 3b is a bulk (not surface) formation and as shown in Fig. 5 is a eutectic mixture (KYF+YF3). However, for KTF crystals (Fig. 3c) it is also a bulk eutectic mixture but of other crystal phases (KTbF4+KTb2F7+KTF) as shown in the manuscript text and described in reference [25] in detail.

Comment 5

- Fig. 4, insert: "Bragg", not "Bregg"

Response 5

We corrected this unfortunate typo in the manuscript.

Comment 6

line 222, "germinated" is not used in crystal growth, use "precipitated" or "segregated" instead

Response 6

Thank you for this remark. We have replaced this term with a more appropriate one.

Comment 7

- in section 3.3, I encourage the authors to discuss the impact of structural defects on the thermal conductivity. While there might be of course a dependence on the crystallographic structure and phonon frequency, the "glass-like" behavior of Na-R-F compounds cannot be explained just with such considerations. Could you estimate the temperature-dependent behavior with simple equations for defect and phonon scattering to see whether the KYF (with its maximum obviously lower than the one for KTF) should have a lower defect scattering or not? The strong difference between KTF and KYF is not really explained by the different masses of Tb and Y in respect to K, except if vacancies were present.

Response 7

It is the structural disordering of the Na-R-F compounds, which is written about in this manuscript that is the main reason for the "glassy" behavior of their thermal conductivity [see ref. 60]. The reasons for this are as follows: indeed, this type of crystals has the following structural features [ref. 69, 70]: fluorine vacancies; additional fluorine ions, shifted along the axes of the second and third orders; quantitative ratio 40/60 between cations of Na and Y differing in size and mass; splitting of some of the cationic positions is possible. In addition, these crystals are characterized by a high degree of anharmonicity of thermal vibrations. All these factors are the reasons for the rich phonon scattering spectrum and the corresponding decrease in thermal conductivity.

And it is difficult to agree with the remark of the reviewer that “…cannot be explained just with such considerations”. The data on the ionic conductivity of these crystals also indicate this [Fedorov, P.P., Sorokin, N.I. & Popov, P.A. Inverse correlation between the ionic and thermal conductivities of single crystals of M1–x R x F2 + x (M = Ca, Ba; R—rare-earth element) fluorite solid solutions. Inorg Mater 53, 626–632 (2017). https://doi.org/10.1134/S0020168517060036]

About estimate of the behavior of the dependence temperature-dependent for defect and phonon scattering the following points can be noted:

It is not possible to propose any simple equation for comparing phonon scattering in KYF and KTF. The differences in the density of KYF and KTF allow us to estimate the difference in the phonon velocity by 15-20%. This gives 15-20% lower thermal conductivity of KTF over the entire temperature range. Differences in the temperature behavior of another characteristic that directly determines the value of thermal conductivity - heat capacity - cannot be very significant. The k (T) dependence for KTF (the smallness of the low-temperature thermal conductivity and its weak temperature dependence) unambiguously indicates the manifestation of phonon scattering. If we compare the ionic conductivity of KYF and KTF, then we can conclude about a more significant manifestation of phonon-defect scattering in KTF (see, for example, Fedorov, P.P., Sorokin, N.I. & Popov, P.A. Inverse correlation between the ionic and thermal conductivities of single crystals of M1–x R x F2 + x (M = Ca, Ba; R—rare-earth element) fluorite solid solutions. Inorg Mater 53, 626–632 (2017). https://doi.org/10.1134/S0020168517060036]/

Comment 8

- why is the ionic conductivity of KYF so much higher than the one for KTF? Again, could vacancies play a role here?

Response 8

The following addition was made to the text of the article: «The ionic conductivity of KYF much higher than the one for KTF, because KYF has a lower potential barrier for charge carrier migration (1.16 eV) than the one for KTF (1.57 eV).». It is rather difficult to say anything definite about the effect of vacancies on ionic transport in KR3F10 crystals. Electrical conductivity is an integral value over all charge carriers.

Comment 9

- what could be the change in ionic transport observed in the Na-R-F- compounds above 750 K?

Response 9

The temperature sdc(T)-dependences for crystals Na0.5–xR0.5+xF2+2x (R = Tb, Y) consist of two linear segments with the temperature of the transition between them T ~750 K (blue and black curves in Fig. 10 b). The each segment of conduction dependences satisfies Arrhenius-Frenkel equation. A similar situation is valid for fluorite-type Ca1-xRxF2+x (R = La-Lu, Y) crystals. [ref. 71]. The specific features common for crystals of Na0.5–xR0.5+xF2+2x and Ca1-xRxF2+x x is the fulfillment of the condition Hs2 > Hs1 , where subscript “1” and “2” refers to a low-temperature region and a high-temperature region respectively. The high-temperature region of conduction is probably connected with the process of dissociation of bonded fluorine ions from clusters [Sorokin, N.I. Superionic transport in solid fluoride solutions with a fluorite structure. Russ. J. Electrochem. 2006, 42, 744-759. https://doi.org/10.1134/S1023193506070081, Archer, J.A.; Chadwick, A.V.; Jack, I.R.; Zeqiri, B. Ionic conductivity studies of heavily rare earth doped fluorites. Solid State Ionics. 1983, 9-10, 505-510 https://doi.org/10.1016/0167-2738(83)90285-0].
